# Accounting for Economic Factors in Socio-Hydrology: Optimization under Uncertainty and Climate Change

**Luis M. Abadie** [1] [iD], **Anil Markandya** [1,2,*] [iD] **and Marc B. Neumann** [1,2]

[1] Basque Centre for Climate Change BC3, Sede Building 1, 1st floor, Scientific Campus of the University of the Basque Country, 48940 Leioa, Spain; lm.abadie@bc3research.org (L.M.A.); marc.neumann@bc3research.org (M.B.N.)

[2] IKERBASQUE, Basque Foundation for Science, María Díaz Haroko Kalea 3, 48013 Bilbao, Bizkaia, Spain

[*] Correspondence: anil.markandya@bc3research.org; Tel.: +34-94-401-46-90 (ext. 117)

**Abstract:** This paper reframes the socio-hydrology analysis as an optimization problem. To achieve this, we first develop a valuation scheme to estimate net benefits of development in a flood plain, consisting of benefits obtained from land and housing, less the costs of flood management and flood damage. Then we look for an optimal safety factor for the levee heightening strategy within the 'Technosociety' scenario for a given time series of future flood events. This is further extended to finding an optimal strategy in the case of uncertainty concerning flood timings and intensities. We suggest an approach for both stationary and non-stationary evolution of flood dynamics and examine how the levee heightening strategy is affected by the magnitude of climate change. We find that the preferred management option depends strongly on the value of the land services the area provides.

**Keywords:** river flooding; extreme events; generalized extreme value distribution; non-stationarity; optimization under uncertainty

---

## 1. Introduction

In an important study [1], the authors examined the feedback between physical and social processes when modeling the management of flood risk. They argued that two important effects determine the social response to floods: the adaptation effect and the levee effect. The former relates to actions taken after a flood event so that the consequences of future events are reduced while the latter relates to the observation that when floods have not occurred for some time, sites become more vulnerable to such events when they do occur, possibly owing to societies taking more risks in their land use policies.

Their novel approach models the links between physical and social processes by comparing two regimes for a flood prone area where settlement has taken place. One regime is referred to as a 'greensociety', where only non-structural measures are used to control floods and the other is called 'technological society', where measures such as levees and dykes form the mainstay of the management of flood risk. In both cases the experience of a flood is assumed to create a memory of the event, to which the community responds, either by reducing population density or heightening the levees. Over time, this effect declines as memory fades and increasing density in the floodplain resumes.

The approach is applied to a situation where there is an increase in the impact of flood frequency and flood magnitude (possibly caused by climate change). Plausible parameters based on the literature are used to track losses, measured in terms of the reduction of population, as well as levee heights for a given schedule of flood events of given magnitude over a 200-year period [1]. The model application shows that the 'greensociety' experiences more frequent flood losses but these are relatively low, while

the technological society experiences high flood losses after a long period of time with non-occurrence of events.

The paper [1] offers many interesting possibilities for extension, and therefore, in this paper, we modify and develop the earlier analysis in the following ways:

1.  In the original paper the costs of each regime are not fully accounted for. Losses from floods include not only population changes, but also loss of physical assets and costs associated with levee construction. In our model these costs are included.
2.  In [1] the model for the technosociety case is myopic. Society reacts only after an extreme event without considering the probabilities (frequency and intensity) of future extreme events. Our model makes use of the calculated probabilities in infrastructure decision-making.
3.  The modeling in [1] is non-stochastic—the sequence of flood events is pre-determined over the century, with 35 extreme events occurring over a 200-year period. A natural extension, therefore, is to explore how the results are affected when the sequence and intensity of extreme events are stochastic in nature requiring optimization under uncertainty to find optimal levee heightening strategies. This is done in our model.
4.  In the framework with stochastic extreme events, we build an objective decision function based on the net present value. This function allows us to obtain expected present values under uncertainty, depending on the adaptation strategy and the model´s stochastic parameters with two risk factors: frequency and intensity of extreme flood events.
5.  Furthermore, we include a model to calibrate the stochastic parameters based on historical information. Using this net-present-value framework, optimal strategies are explored, e.g., an optimal levee heightening strategy, which could not be done in the original model.

To some extent, these aspects have been analyzed in other recent literature on socio-hydrology. The economic factors relating to floods have been studied in this context by the authors of [2], seen in their approach models, the choices between investing in flood defenses and in productive capital at the national level. Higher investment in flood protection reduces damages, which in turn affects the productivity of the economy and thereby the level of welfare. The objective is to determine the optimal path for investment in the two types of capital in a non-stochastic setting, where optimality is defined as maximizing the discounted present value of consumption over an infinite horizon. The authors find that the optimal problem has multiple solutions and the one attained depends on the initial conditions in terms of capital stock. Our analysis is different from theirs in two respects. First we do not seek an optimal solution in an economy-wide setting. Such a solution is interesting in understanding the broad trade-offs but less useful in determining the consequences of decisions at the local level, which are usually not taken in an economy-wide optimizing context. Indeed, one of the strengths of the socio-hydrology approach is to stress how actual decisions reflect recent memory and how they respond to perceived 'safety' by over-development in the floodplain. Our economic analysis seeks to understand what the actual outcome will be in economic terms if one or the other of the social rules are followed in flood management. Second, we seek to understand how to manage optimally some of the exogenous parameters that determine the economic outcome in a local setting, given that actual decision rules are what they are.

In [3], the authors introduce stochasticity into the socio-hydrological analysis by considering a Poisson distribution for the number of occurrences per unit time and a generalized Pareto distribution for the magnitude of high water levels. The study investigates the evolution of settlement size and flood damage evolution with respect to parameters associated with trust, collective memory and risk-taking attitude. However, the authors do not include considerations of optimization. Here we try to find an optimal strategy under uncertainty considering both stationary and non-stationary climatic processes.

In [4], the analysis is based on hydraulic modeling of historical events and the authors show that human interventions on both the landscape and the subsoil have altered the flood dynamics, increasing hydraulic hazard.

In an analysis of flood risk change [5] in the floodplain of the Emme River downstream of Burgdorf, Switzerland, the authors show that the construction of lateral levees and the river incision following its construction are the main drivers for decreasing flood risks over the last century. The authors state that a rebound effect due to settlement growth after levee construction will become increasingly relevant in the future with continued socio-economic growth.

The relationship between long-term changes in human proximity to rivers and the occurrence of catastrophic flood events was analyzed by the authors of [6], including how these relationships are influenced by different levels of structural flood protection. The authors found that societies with low protection levels react to flood events by resettling further away from the river, while societies with high protection levels show no significant changes. Indeed, they continue to rely heavily on structural measures, reinforcing flood protection and quickly resettling in these flood-prone areas.

To develop our study, we additionally draw on the following works. A review of flood risk literature can be found in [7]. These authors identified 258 articles addressing governance and flooding, resilience and adaptation.

Using the district of Maxvorstadt in Munich for demonstration, the authors of [8] introduce a time-varying flood resilience index (FRI) to quantify the resilience level of households.

The public flood risk perception in four districts of Jingdezhen is analyzed by [9], examining the influencing factors.

The time series of floods across the Niger River basin was analyzed by the authors of [10]. These authors found an increasing number of catastrophic floods with the most extreme increase in the Middle Niger.

A study of Pan-European river flow was realized by the authors of [11], using simulations coupled with a high-resolution impact assessment framework based on a 2D inundation model, using two methods. Their event-based work includes changes in time of hazard, exposure and vulnerability. Their integral method reproduces the average flood losses which occurred in Europe between 1998 and 2009.

A sensitivity analysis, which considers changes in all risk components such as changes in climate, catchment, river system, land use, assets, and vulnerability was made by the authors of [12] for the mesoscale Mulde catchment in Germany, showing that flood risk can vary dramatically as a consequence of different scenarios.

In this paper, we compare a 'greensociety' (no levees) with a 'Technosociety' that seeks an optimal levee heightening strategy with respect to the net present value of economic benefits for both deterministic and stochastic patterns of extreme events.

Section 2 develops the methodology laying out some economic parameters for a floodplain under development during two centuries, looking at two cases: one where the initial land and property values are relatively high and one where they are relatively low. It also considers flood costs, and dyke costs for two scenarios following the flood event: only extension of levees or complete reconstruction with extension. Section 3 presents and discusses the results looking for the optimum levee heightening strategy that leads to minimum net costs for a given deterministic pattern of extreme events. This section also extends the approach for a stochastic climate signal where flood events occur as a Poisson process and their intensity is modeled using a generalized Pareto distribution. The analysis is conducted for a stationary as well as a non-stationary climate signal; for the second case we also analyze how the strategy changes as a function of climate change intensity. Section 4 concludes and lays out how the analysis here extends the earlier analysis in socio-hydrology and possible further steps in research.

The research presented in this paper should be of use to authorities responsible for the management of land use in flood plains in different countries. The purpose of a socio-hydrology approach is to recognize the limited rationality in reactions after flood events, and the way in which memory fades and land use reverts to patterns that ignore the lessons from history. The model here looks at how overall benefits from the land area can depend on different rules of thumb—one being a more technological one and one being a ´greener´ one. The rule that works best depends on some basic parameters of the

system, which are investigated here. The paper also determines the optimal levee height strengthening strategy under the technological model. It turns out that land values and their expected increases are critical factors in determining the choice of a rule for management. This will apply equally in developing and developed countries, where flood plains can have very varied land values.

## 2. Methodology

The net costs over a period of time under a given management regime are the costs from flood damages, plus the costs of flood control less, the services received from the land in the flood plain. The sum can be written as $\psi$ in Equation (1):

$$\psi = \sum \frac{V(t)}{(1+r)^t} \tag{1}$$

$$V(t) = \pi_1(t)F(t) + C(t) - L(t) \tag{2}$$

where $F$ is the damage caused by the flood and $\pi_1$ is the value of a unit flood event and $C(t)$ is the cost of the defenses. $L(t)$ is the value of land and housing services in the floodplain. The aim for a management regime is to minimize the value of the function $\psi$. Each component of (2) is described below.

### 2.1. Land Services

The land services can be assumed to be a function of the number of people living in the floodplain, measured through the rent on the land they occupy and use. As the number increases, so does the value of the services, but the relationship is not straightforward. An increase in population with a fixed supply of land increases density and the price of land. Studies indicate that the relationship is "U" shaped, with the rate of increase of price declining and reaching a maximum before it starts to fall [13]. Using Polish data, they estimate the coefficient of density (measured as flood area ratio (FAR)) on the log of price to be around 0.55. But they also cite a Japanese study [14] showing that the price peaks with a density of FAR of between 1.1 and 1.7.

Given the initial values for density in [1], we have constructed two baseline scenarios, one reflecting areas with relatively high initial land and property values and the other reflecting low initial values (Table 1). The high value combination is considered plausible for a high income country while the low value combination is considered plausible for a low income country at the present time.

**Table 1.** Baseline initial values for land services for high and low value scenario.

| Variable | Unit | High Value | Low Value | Basis |
|---|---|---|---|---|
| Initial Population | No. | 25,000 | 1000 | Assumed |
| Area ($A$) | Hectares | 100,000 | 10,000 | Assumed |
| Residential Use Share ($\mu$) | % | 40 | 40 | Common for such use |
| Pop. Density ($D$) | Pop/Ha | 0.25 | 0.10 | Calculated |
| Dwelling Density ($\delta$) | Dwelling/Ha. | 0.10 | 0.04 | Calculated |
| FAR | Ratio | 0.0025 | 0.0010 | Plausible values |
| Value of Land | USD/Ha. | 100 | 100 | Assumed |
| Value per Dwelling ($B$) | USD | 100,000 | 4683 | Plausible range for low and high income countries |
| Value of all Land with Dwellings | USD Mn. | 1004.0 | 2.3 | Calculated |

In order to show how land values change over time, a quadratic Equation (3) is used:

$$Ln(P) = a + bFAR + cFAR^2 \qquad (3)$$

where *P* is the price of land, *FAR* the floor area to the plot area and a, b and c, parameters. From [13], the value of parameter 'b' is taken as 0.55. If the function peaks at a value of d of 1.4 (see above) then one can estimate 'c' as –0.196. Given initial values as specified in the table, the values of 'a' comes out at 4.6 in both cases, (the values are slightly different in the two cases, but the differences are very small).

From the numbers in Table 1 the relationship between FAR and dwelling density D is given in both cases by Equation (4):

$$FAR = 0.025D \qquad (4)$$

This means Equation (3) can be written in terms of *D* as in Equation (5):

$$Ln(P) = 4.6 + 0.01375D + 0.0001225D^2 \qquad (5)$$

In addition, we allow for an increase in the value of land and housing services due to overall growth in the economy and to population growth. Typical rates of per capita economic growth over periods of 100 years are around 2% [15], which is what is taken. Population growth is taken as in [1] as 3%.

The resulting function of land and housing services that emerges from the analysis is given by Equation (6):

$$L(t) = e^{gt} \cdot r \cdot \mu \cdot A \cdot P(t) \left\{ 1 + \left( \frac{B \cdot \delta}{P(0)} \right) \right\} \qquad (6)$$

The unit value in parenthesis multiplied by the term outside the parenthesis gives the value of land services and the second term the value of housing services. The 3% increase in density in [1] translates into a 3% increase in population, as the land area is fixed. Hence the proposed value of g is 0.05 (2% for per capita growth and 3% for population) and the value of *r* (the discount rate) is 0.03 (i.e., 3%). The values of *μ*, *A* and *B* are as given in Table 1. *P(t)* is the price of land at time *t*, and *P(0)* is the price at time *0*.

## 2.2. Losses in the Floodplain from Floods

The flood losses for any one event are set in the unit interval in the original paper and we keep the same range. A value of 1, which is the maximum, would mean total loss of all property (i.e., housing assets but not, of course, the land). The value of that loss will depend on the year in which it happens and the number of dwellings. If we assume no loss of life the value of the maximum loss in year (t) is given by Equation (7):

$$\pi_{1(t)} = e^{gt} \cdot \mu \cdot A \cdot \frac{P(t)}{P(0)} B \cdot \delta. \qquad (7)$$

## 2.3. Costs of Raising and Replacing Flood Defenses

After a flood event the dam height is increased from H1 to H2. We include the possibility that a fraction υ of dams need to be rebuilt and consider the extreme cases of υ = 0 (only dam heightening required) and υ = 1 (complete rebuilding including heightening). We consider a simplified schematic of flood plain geometry and dam geometry (Figure 1).

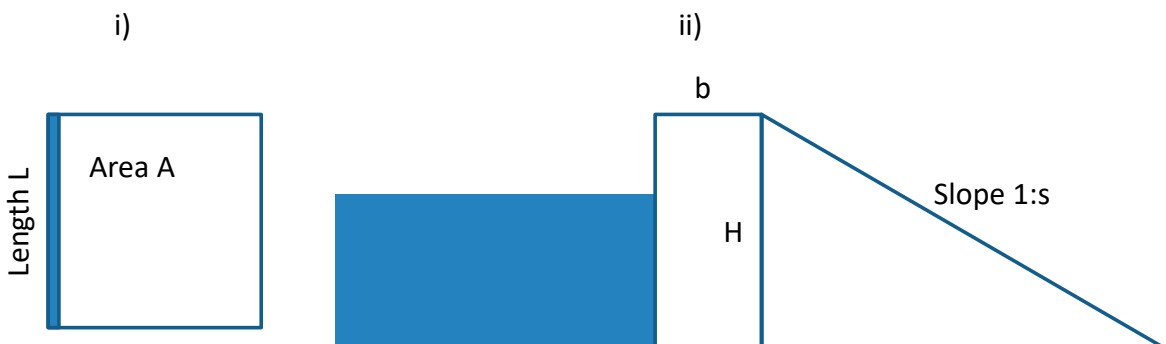

**Figure 1.** Schematic for river in flood plain and dam: i) The river runs along the side of a square representing the flood plain with area *A*. ii) Dam geometry with crest b, height H and slope 1:*s*.

The cost of dam heightening and replacement following a flood event is then given by Equation (8):

$$C(t) = kL\left\{(1-v)\left[\left(bH_2 + \frac{sH_2{}^2}{2}\right) - \left(bH_1 + \frac{sH_1{}^2}{2}\right)\right] + v\left[bH_2 + \frac{sH_2{}^2}{2}\right]\right\} \tag{8}$$

where
    *k* (unit cost of dam construction, assumed constant at *k* = 50 USD/m$^3$)
    *L* (length of dam, *L* = *A*$^{0.5}$)
    $v$ (faction of dams that need to be rebuilt, $v$ = {0,1})
    *b* (weir crest, *b* = 3 m)
    *s* (inverse of slope 1:*s*, *s* = 3)

The authors are aware that these assumptions reflect a highly simplified system. However, they serve the illustrative purpose of the paper and can be easily adapted to more complex floodplain or dam geometries. For example, in the case of a two sided dam the cost function would need to be multiplied by two. The function could also be extended to unit costs that vary with construction size.

### 2.4. Optimization Methods

### 2.4.1. Optimization Methods for a Deterministic Characterization of Flood Events

In order to apply the model, a number of parameters determining the relative flood damage and the amount by which the levee height is raised need to be fixed. These are set at the same values as in the original paper. The figures are given in Table 2. The optimization application tracks the dynamic path for the evolution of density (*D*), levee height (*H*) and memory (*M*) as given in [1]. For the reader's benefit they are reproduced in the Appendix A.

**Table 2.** Parameters of the model as set in [1].

| Parameters | Description | Domain | Value |
|---|---|---|---|
| $\alpha_H$ | Parameter related to relationship between flood water levels to relative damage | Hydrology | 10 m |
| $\xi_H$ | Proportion of flood level enhancement due to flood levees | Hydrology | 0.2 |
| $\alpha_D$ | Ratio of preparedness/awareness | Demography | 5 |
| $\varepsilon_T$ | Safety factor for levee heightening | Technology | 1.1 |
| $\kappa_T$ | Protection level decay rate | Technology | $2 \times 10^{-5}$ year$^{-1}$ |
| $\mu_S$ | Memory loss rate | Society | 0.06 year$^{-1}$ |

Source: [1].

2.4.2. Optimization Model for a Stochastic Characterization of Flood Events

Given the base case data of 35 extreme events in 200 years with intervals of dt years, the parameter lambda (frequency) is defined as in Equation (9):

$$\lambda = \frac{35}{200}dt \tag{9}$$

When dt = 1, we have λ = 0.175. The model allows, however, the use of time intervals dt < 1, for example dt = 0.10 as is used here.

With the Poisson distribution, the probability of observing k events in an interval is given by Equation (10):

$$e^{-\lambda}\frac{\lambda^k}{k!} \tag{10}$$

The probability of a given number of extreme events occurring in a year can then be calculated (Table 3).

**Table 3.** Number of events in one year under a Poisson process.

| Events (k) | Probability (%) |
| :---: | :---: |
| 0 | 83.946 |
| 1 | 14.690 |
| 2 | 1.285 |
| 3 | 0.075 |
| >3 | 0.003 |
| Total | 100.00 |

For the intensity, we considered two possible distributions for modeling of extreme events: the truncated generalized extreme value (GEV) and the generalized Pareto distributions. The GEV includes, as particular cases, the Gumbel, Fréchet and Weibull distributions and has been developed within the extreme value theory literature (see [16] and [17]). As such, it seems an appropriate choice for extreme event modeling of floods. For example, the authors of [18] use the generalized extreme distribution for sea level rise events. However, it has the disadvantage of taking on negative values, which means that using it in this context would require truncating the distribution to zero.

The generalized Pareto distribution, on the other hand, cannot produce negative values. The cumulative distribution function for a generalized Pareto distribution is in Equation (11):

$$1 - (1 + k\frac{x - \theta}{\sigma})^{-\frac{1}{k}}, k \neq 0 \tag{11}$$

where $\theta \in R$ is the location parameter, $\sigma > 0$ is the scale and $k \in R$ is the shape.

Fitting this distribution to the 35 extreme events in the original dataset, gives the values in Table 4. The histogram of the original events and the density function of the estimated generalized Pareto distribution are shown in Figure 2.

**Table 4.** Parameter values for the generalized Pareto distribution corresponding to the dataset.

| | Parameter Value | 95% Confidence Interval |
| :---: | :---: | :---: |
| Shape ($k$) | −0.466 | −0.761 to −0.172 |
| Scale ($v$) | 9.812 | 6.476 to 14.868 |

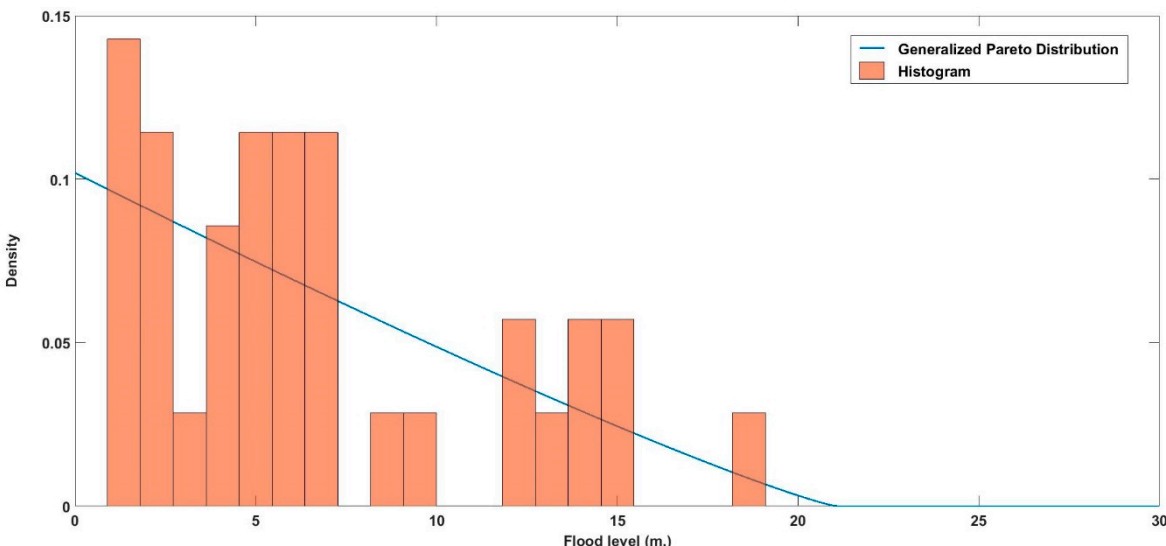

**Figure 2.** Flood level with generalized Pareto distribution.

We obtained goodness of fit measures for both the truncated generalized extreme value distribution ($\chi^2 = 0.845$) and for the generalized Pareto distribution ($\chi^2 = 0.580$). Since the goodness of fit test statistics indicate the distance between the data and the fitted distributions, the generalized Pareto distribution with the lower $\chi^2$ statistic value is the better fitting model and therefore the one used in the following sections

### 2.4.3. Optimization Model for Non-Stationarity Arising from Climate Change

The Intergovernmental Panel on Climate Change (IPCC) reports project a likely increase in both frequency and intensity of extreme climate change events with particularly strong effects on countries already vulnerable to extremes of normal climatic variability. Recent work [19,20] provides estimates of the likely magnitude of these effects. There is, however, considerable uncertainty on both the rapidity and magnitude of climate change and its implications for extreme events. We therefore perform a sensitivity analysis with respect to climate change magnitude for the high land value case. In this subsection we examine the impact of an increase in the expected frequency of extreme events caused by climate change, assuming an exponential increase with time similar to that used to model other climate events such as sea level rise.

We assume that the frequency of extreme events grows exponentially according to Equation (12):

$$\lambda(t) = \lambda(0)e^{\alpha_\lambda t} \tag{12}$$

The expected frequency of extreme events depends on the year and grows exponentially at the rate $\alpha_\lambda$.

The generalized Pareto distribution mean, when $\theta = 0$, is expressed in Equation (13):

$$\frac{\sigma}{1 - k} \tag{13}$$

Further, we also assume an exponential growth of intensity as expressed in Equation (14):

$$\frac{\sigma(t)}{1 - k} = \frac{\sigma(0)e^{\alpha_\sigma t}}{1 - k} \tag{14}$$

## 3. Results and Discussion

### 3.1. Optimization Results for a Deterministic Characterization of Flood Events

The results of applying the model to the two sets of land values and populations described in the Methodology section are given in Table 5. They are in present value US dollars (billions) covering the 200-year period 1900–2100. Two variants are shown: In the case of ($\upsilon = 0$) only extension is required whereas for $\upsilon = 1$ rebuilding with extension is required. The results also depend on the step interval in the calculation. We use a temporal resolution of 0.1 years.

**Table 5.** Cost estimates from the model: deterministic case (USD million).

| | Technosociety | | Greensociety |
|---|---|---|---|
| | **Case of $\upsilon = 0$** | **Case of $\upsilon = 1$** | |
| High Value | 1719 | 1927 | 5158 |
| Low Value | 61 | 127 | 31 |

For the high value case the costs are lower in the 'technosociety' whereas for the low value case they are lower for the 'greensociety'. Thus it appears that under quite a wide range of cost assumptions ($\upsilon = 0$ to 1) a high value area will benefit from a 'technosociety' management regime but a low value area will benefit from a 'greensociety' regime.

To investigate how the regimes might be affected by the choice of parameters, we examine the sensitivity of the results to a key parameter—the safety factor for levee heightening ($\varepsilon_T$). The safety factor $\varepsilon_T$ is the proportional raising of defenses when a flooding events occurs, as presented in the Appendix A Equation (A2). This safety factor in [1] is 1.1 for 'technosociety' and 0 for 'greensociety' (as no levees are present). Here we examine how the outcome for 'technosociety' is affected as a function of the safety factor for the high value and low value cases respectively (Figures 3 and 4). It is interesting to observe the presence of multiple local minima and the non-smooth behavior which are both a reflection of the deterministic model. In the high value case (Figure 3) the optimal value for the safety factor of levee heightening is 3.76, much higher than the value of 1.1, which was established for the 'technosociety' in the original paper. In the low value case, the optimal value is zero, i.e., no levees are built (Figure 4).

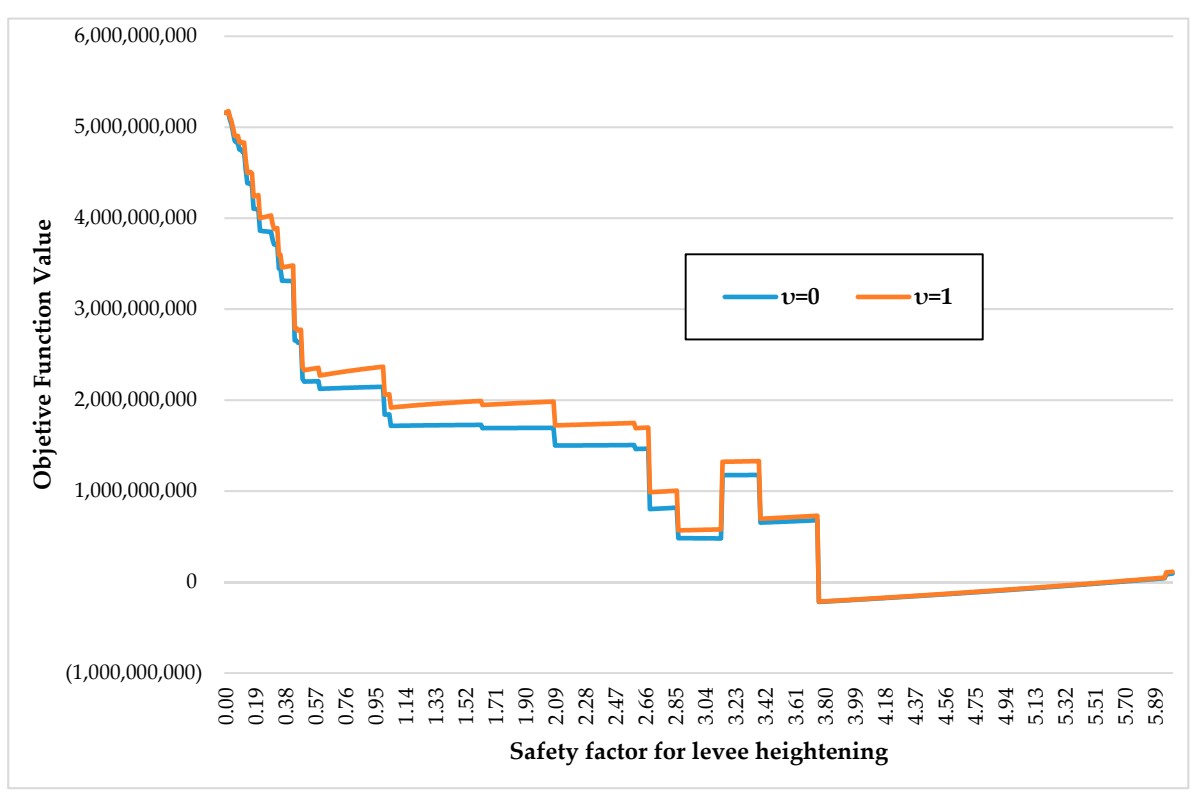

**Figure 3.** Objective function variation with the safety factor (high value case).

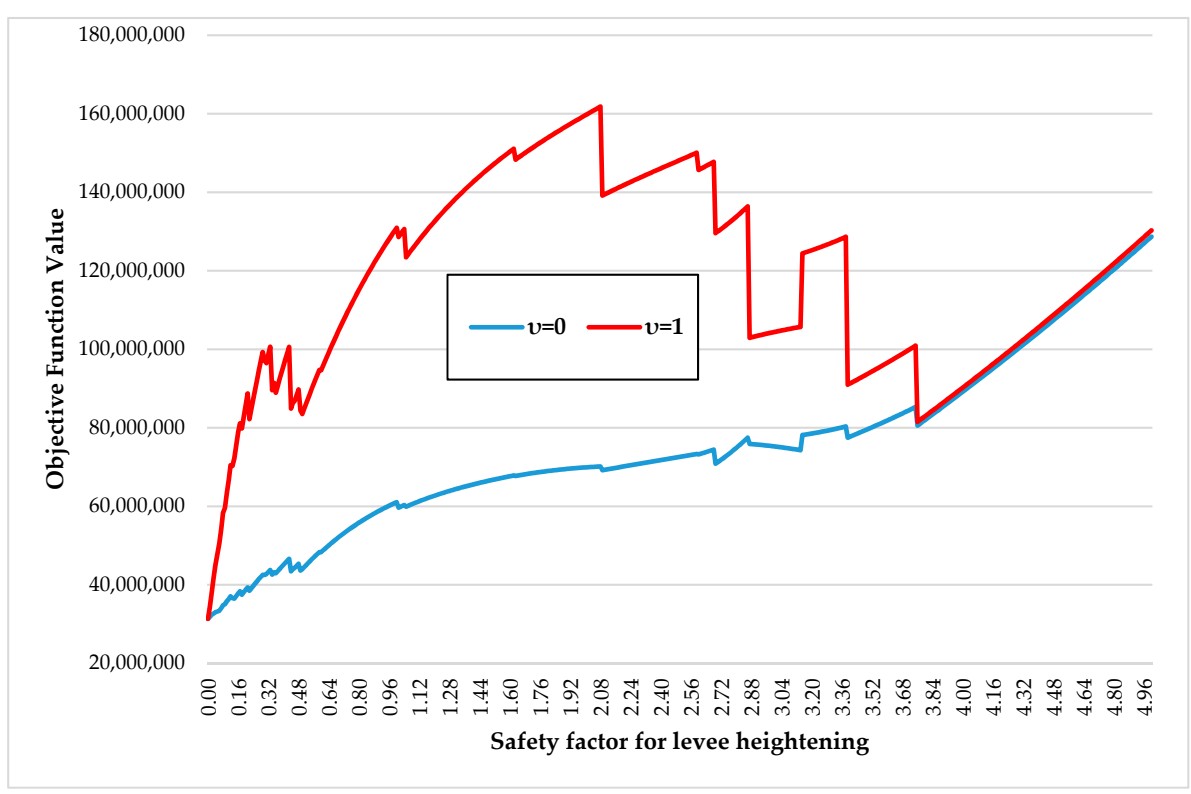

**Figure 4.** Objective function variation with the safety factor (low value case).

*3.2. Optimization Results for a Stochastic Characterization of Flood Events*

For the simulations we take dt to be 0.10 with the generalized Pareto shown in Figure 2. A total of 50,000 paths were simulated using random numbers from the Poisson distribution for the parameter

lambda. Each path has 2000 (200 years divided by dt) values (zero or one). We then extract a matrix of 2000 by 50,000 cells for the generalized Pareto distribution. The simulated values in the generalized Pareto case have a mean intensity of 6.76.

Each of the 50,000 paths is processed to obtain 50,000 values of the objective function at $\varepsilon_T = 1.1$. The mean of these values is the expected present value under uncertainty, as all paths have the same probability. Figures 5 and 6 present the high land value scenario for the 'technosociety' and 'greensociety' whereas Figures 7 and 8 do the same for the low value scenario.

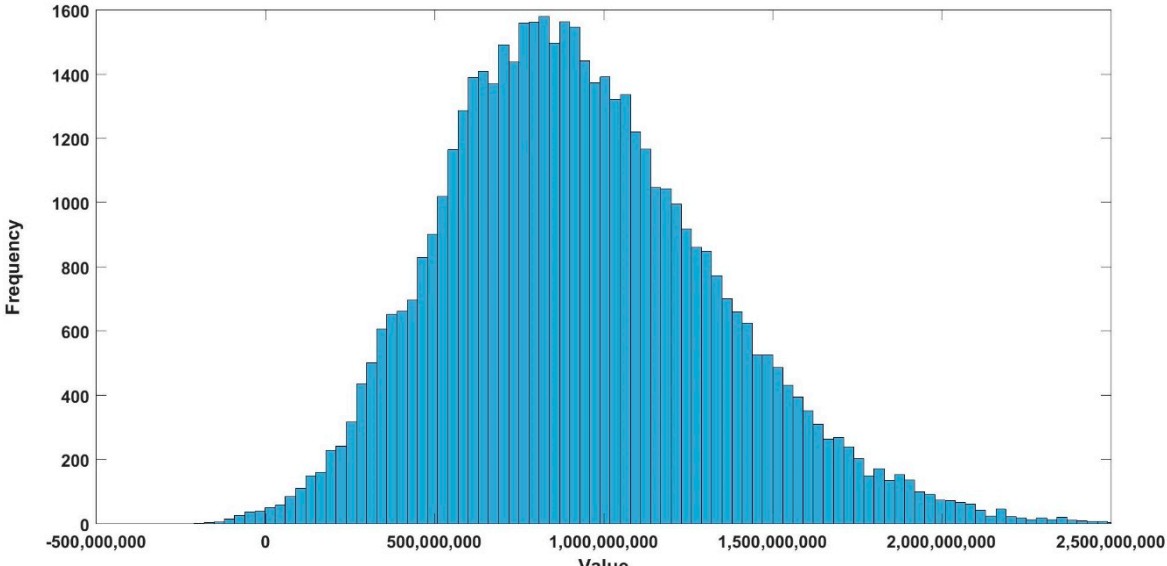

**Figure 5.** Histogram of the objective function at $\varepsilon_T = 1.1$ for the technosociety/high value case with $\upsilon = 0$.

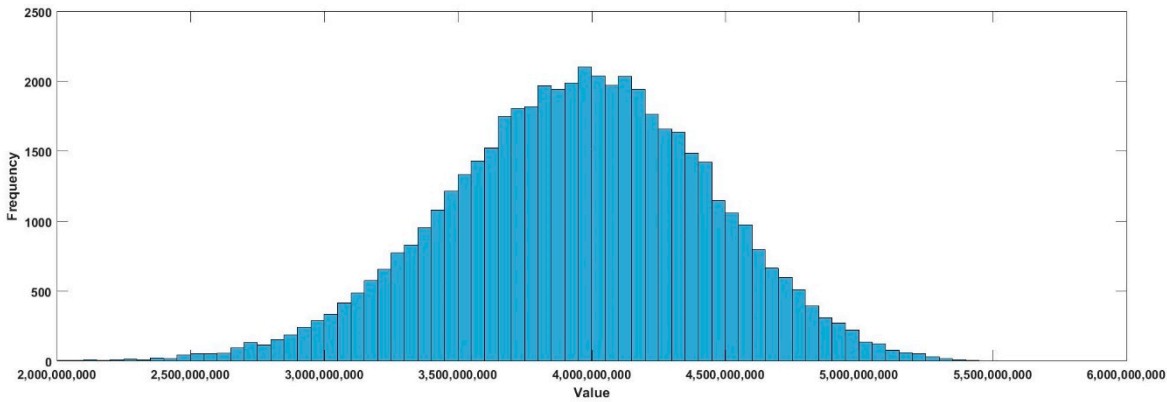

**Figure 6.** Histogram of the objective function for the greensociety/high value model.

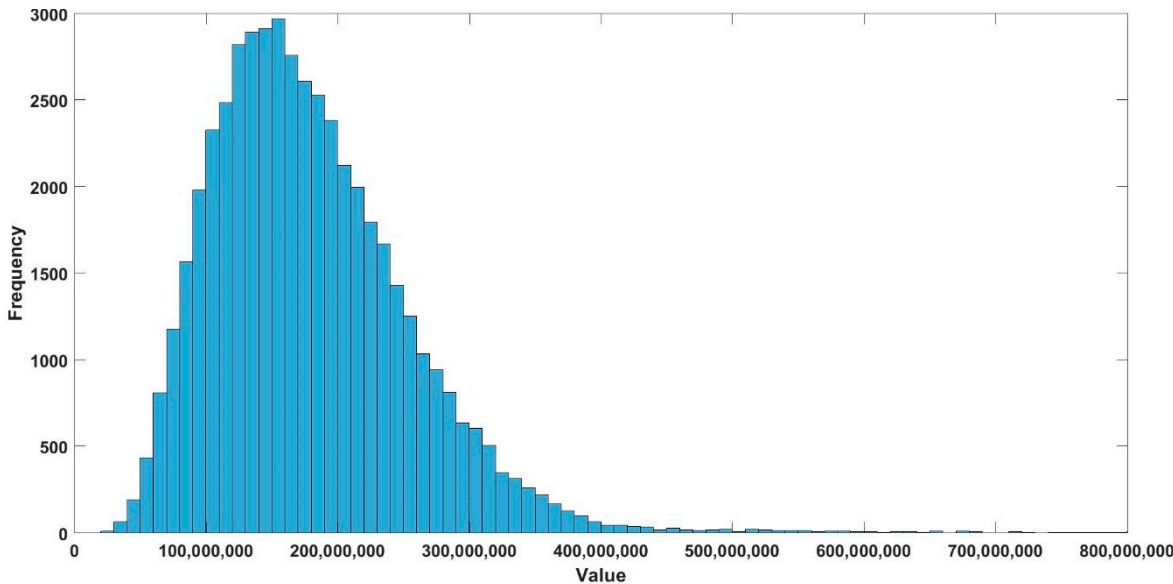

**Figure 7.** Histogram of the objective function at $\varepsilon_T = 1.1$ for the technosociety/low value case with $\upsilon = 0$.

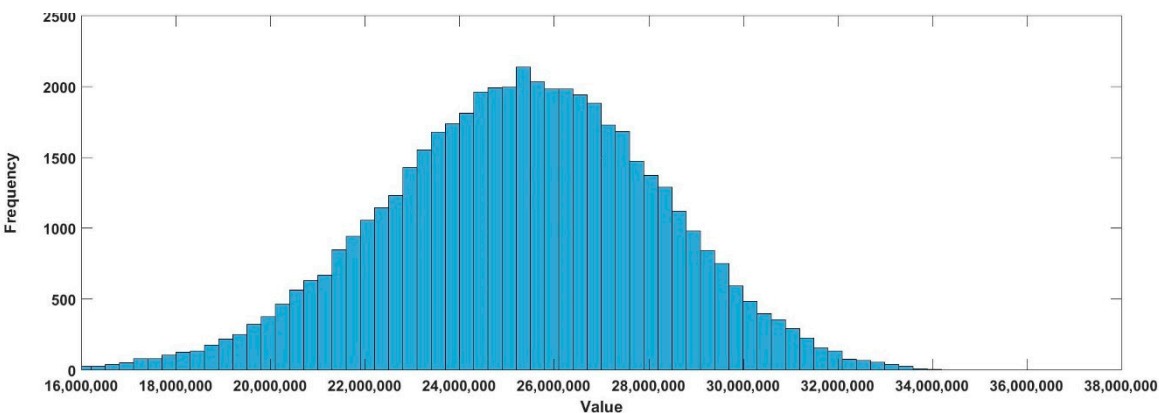

**Figure 8.** Histogram of the objective function for the greensociety/low value case.

Table 6 summarizes the results in terms of the expected net present value (NPV) for the two values of $\upsilon$ and with the values of safety factor ($\varepsilon_T$) of 1.1 for 'technosociety' and 0 for 'greensociety' (as in the original paper). The results have the same ordering in terms of cost as the deterministic case: the 'technosociety' option has a lower cost in the high value case and the 'greensociety' has the lower cost in the low value case.

**Table 6.** Expected net present value (NPV) (million) for the different management regimes in the stochastic case.

| Generalized Pareto | High Land Value Scenario | | Low Land Value Scenario | |
|---|---|---|---|---|
| | **Technosociety** | **Greensociety** | **Technosociety** | **Greensociety** |
| $\upsilon = 0$ | 993 | | 181 | |
| $\upsilon = 1$ | 1290 | 3947 | 294 | 25 |

Corresponding to the optimization with respect to the safety factor in the deterministic case, we now seek to minimize the objective function with respect to the safety factor in the stochastic case. This is done by repeating the analysis above and extracting the expected values from the distributions obtained at different values for the safety factor. Figure 9 shows the high land value scenario values under uncertainty as a function of $\varepsilon_T$ and Figure 10 does the same for the low land value scenario.

In both cases $\upsilon = 0$. Two observations can be made. The functions are smooth in comparison to the case discussed above when optimizing in the presence of a deterministic signal. Also, the minimum in Figure 9 is quite flat, implying that under uncertainty safety factors, between 1 and 2 lead to similar economic outcomes.

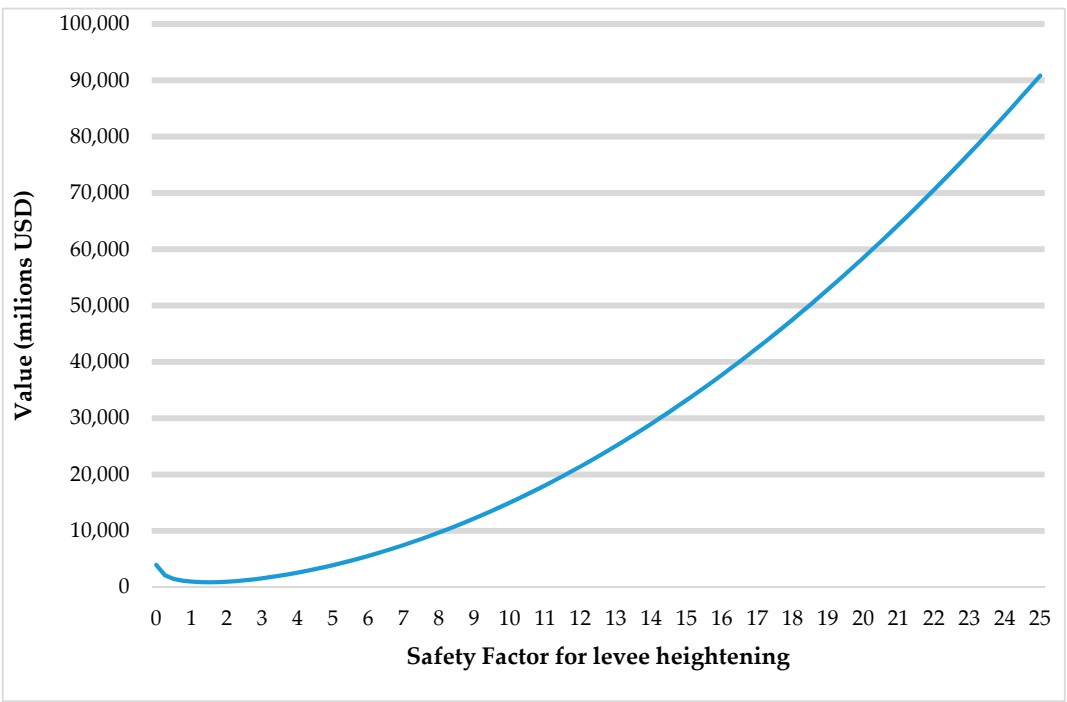

**Figure 9.** Objective function for the high value case depending on $\varepsilon_T$ when $\upsilon = 0$.

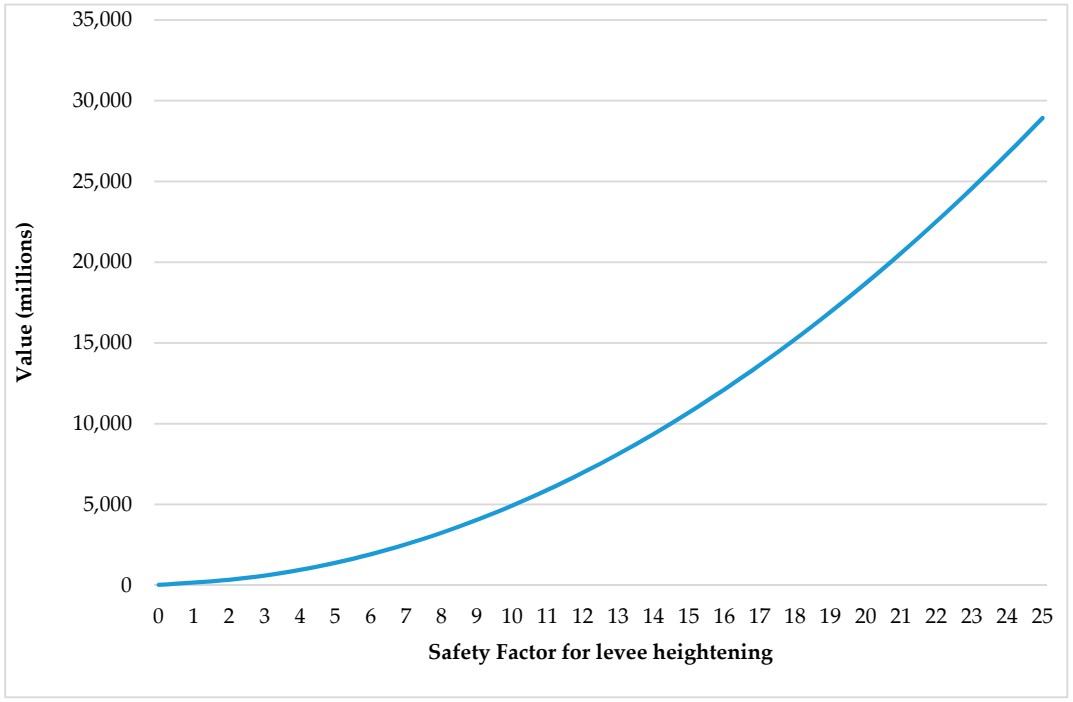

**Figure 10.** Objective function for the low value case depending on $\varepsilon_T$ when $\upsilon = 0$.

Table 7 shows that the optimal value of $\varepsilon_T$ in the high land value scenario is 1.53 when $\upsilon = 0$ and 1.64 when $\upsilon = 1$ when assuming the generalized Pareto distribution. For the truncated GEV case the

respective values are 1.17 and 1.24. Hence, the desired safety factor is not very sensitive to the value of $\upsilon$. In the low land value scenario the desired safety factor is zero for both types of distribution.

**Table 7.** Optimal safety factor for levee heightening for Technosociety.

| Generalized Pareto | High Land Value | Low Land Value |
|---|---|---|
| $\upsilon = 0$ | 1.53 | 0.00 |
| $\upsilon = 1$ | 1.64 | 0.00 |

### 3.3. Optimization Results for Non-Stationarity Arising from Climate Change

We perform Montecarlo simulations with Equations (14) and (16). Using $\alpha_\lambda = \alpha_\sigma$ values of 0.001, 0.002, 0.003, 0.004 and 0.005, we obtain the optimal decision values for levee heightening of Table 8.

**Table 8.** Impact of climate change for the high value case. Upper section: multipliers for frequency and intensity changes for the 100 and 200-year time horizon respectively. Lower section: optimal levee heightening strategy (for two cases: $\upsilon = 0$: only extension required after event, $\upsilon = 1$: rebuilding and extension required after event).

| $\alpha_\lambda = \alpha_\sigma$ | 0.000 | 0.001 | 0.002 | 0.003 | 0.004 | 0.005 |
|---|---|---|---|---|---|---|
| $e^{\alpha_\lambda t}$ | | | | | | |
| 100 years | 1.00 | 1.11 | 1.22 | 1.35 | 1.49 | 1.65 |
| 200 years | 1.00 | 1.22 | 1.49 | 1.82 | 2.23 | 2.72 |
| Safety Factor for levee heightening $\varepsilon_T$ | | | | | | |
| $\upsilon = 0$ | 1.53 | 1.50 | 1.45 | 1.35 | 1.27 | 1.17 |
| $\upsilon = 1$ | 1.64 | 1.67 | 1.70 | 1.71 | 1.65 | 1.64 |

High-end climate change scenarios imply higher levees, but, surprisingly, not necessarily higher safety factors (Table 8 and Figure 11).

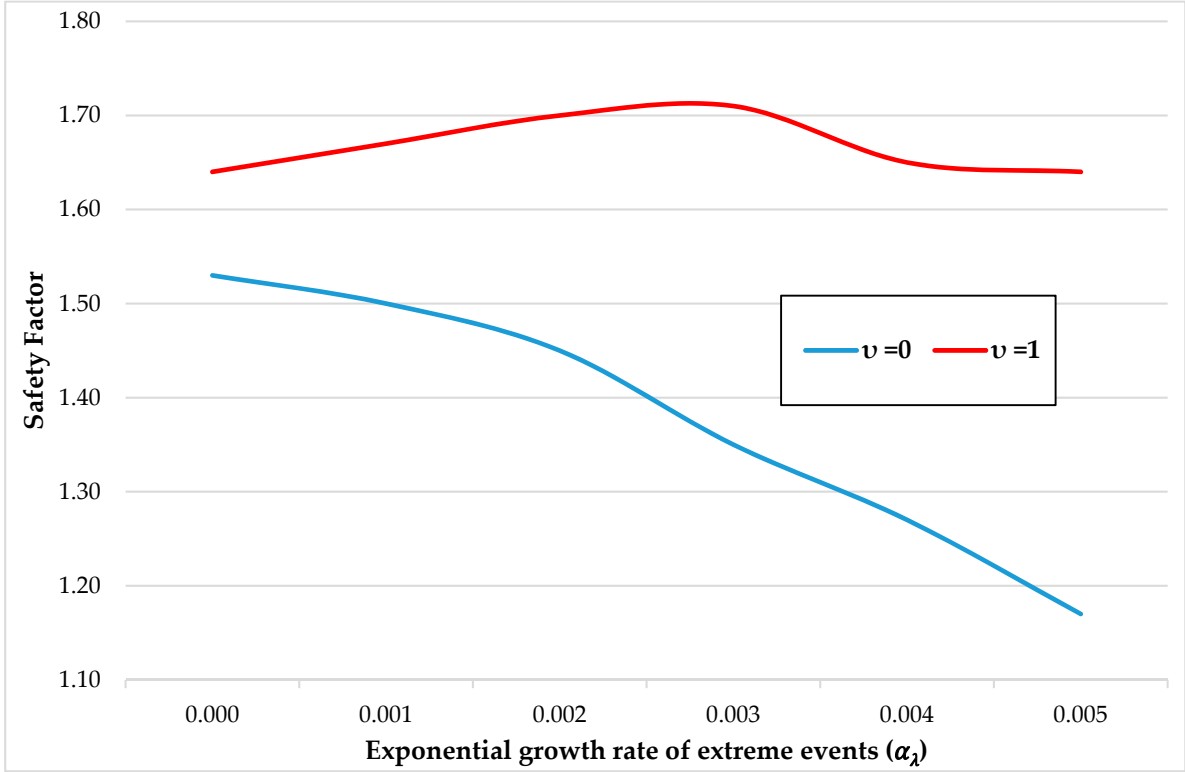

**Figure 11.** Optimal $\varepsilon_T$ values as a function of growth rate of extreme events.

In the future, as knowledge improves about the actual climate pathway, the calculations can be redone to obtain revised safety factors for future investments.

## 4. Conclusions

This paper shows how the outcomes of the socio-hydrology problem described in [1] can be reframed as an optimization problem, by including an economic valuation of costs and benefits. It contributes the following to what this earlier paper had shown:

1. The better management option ('technosociety' or 'greensociety') depends on the initial value of the land services the area provides and the evolution of those services over time. We have plausible cases of high initial land values where the 'technosociety' option is better and of low initial land values where the 'greensociety' is better.
2. The paper shows that a critical parameter for the values under the 'technosociety' in determining the values of the flood plain is the safety factor $(\varepsilon_T)$ (i.e., the proportional raising of defenses when a flooding events occurs). For the deterministic case considered in the original paper, we obtain an optimal safety factor that is significantly higher than in the original paper.
3. Whereas [1] consider only myopic behavior toward flood events, our model calculates the value of land services and other costs under uncertainty of extreme event occurrence and intensity. The paper shows the important differences in results when optimizing either under certainty or uncertainty. Our approach allows the economically optimal levee heightening strategy to be calculated under uncertainty.
4. We show how climate change affects the optimal response to floods in a socio-hydrology framework. It turns out that the greater the expected impacts from such change the higher the levees but, surprisingly, the optimal safety factors do not always increase.

A further possible extension is to explore how the two regimes can be modified to yield higher benefits. This may be done, for example, by incentives in the form of taxes or subsidies on levee construction. A non-trivial exercise would be required to explore how the behavior-giving equations would need to be adapted to adequately mimic the response of society towards such incentives.

**Author Contributions:** A.M. and M.B.N proposed the work; L.M.A. developed the software for calculation; L.M.A., A.M. and M.B.N. analyzed the results and wrote the paper.

**Funding:** This research is supported by the Basque Government through the BERC 2018-2021 program and by the Spanish Ministry of Economy and Competitiveness MINECO through the BC3 María de Maeztu excellence accreditation MDM-2017-0714. Luis Mª Abadie is grateful for the financial support received from the Spanish Ministry of Economy and Competitiveness (MINECO RTI2018-093352-B-I00). Marc B. Neumann acknowledges funding (RYC-2013-13628) obtained from the Spanish Ministry of Economy and Competitiveness (MINECO).

**Conflicts of Interest:** The authors declare no conflict of interest.

## Appendix A

In this appendix, we show the differential equations for Di Baldassarre et al. (2015) [1] used in the calculations.

The damage caused by flood is as in Equation (A1):

$$F(t) = 1 - e^{-\frac{W(t) + \xi_H H_-(t)}{\alpha_H}} \; if \; W(t) + \xi_H H_-(t) > H_-(t) \tag{A1}$$

where *W(t)* is the flood level without defenses, H is the defenses height. The sub-indices refers to previous moments.

The amount *R* by which the defenses are raised in meters is presented in Equation (A2):

$$R = \varepsilon_T[W(t) + \xi_H H_-(t) - H_-(t)] \tag{A2}$$

With the values of Table A1.

**Table A1.** Society and safety factor for levee heightening.

| $\varepsilon_T$ | Society |
|:---:|:---:|
| 1.1 | Technosociety |
| 0.0 | Greensociety |

The Equation (A3) is the memory equation:

$$\frac{dM}{dt} = \Delta(\psi(t)) \times F \times D_- - \mu_S M \tag{A3}$$

The height levees evolution differential equation is modeled with the Equation (A4)

$$\frac{dH}{dt} = \Delta(\psi(t))R - \kappa_T H \tag{A4}$$

And the densinty differential equation is the Equation (A5)

$$\frac{dD}{dt} = \rho_D - \rho_D D - \rho_D \alpha_D D \times M - \Delta(\psi(t)) \times F \times D_- \tag{A5}$$

If memory $M = 0$ and no flooding events $\Delta(\psi(t))$ then we have:

$$\frac{\delta D}{\delta t} = \rho_D - \rho_D D \tag{A6}$$

We can demostrate than the solution of this equation is:

$$D(t) = 1 + (D(0) - 1)e^{-\rho_D t} \tag{A7}$$

This value is between zero an one.

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
