# Peer review of "Accounting for Economic Factors in Socio-Hydrology: Optimization under Uncertainty and Climate Change"

_water, doi:10.3390/w11102073_

Round 1

Reviewer 1 Report

The paper results well written and interesting.

the study reframes a previous study as an optimization problem developing at first a valuation scheme to estimate net benefits of development in a flood plain and then looking for an optimal safety factor for the levee heightening strategy for a given time series of future flood events. The study then examines how the levee heightening strategy is affected by the magnitude of climate change.

Few comments are listed in the attached document.

Author Response

REVIEWER 1

The study reframes a previous study as an optimization problem developing at first a valuation scheme to estimate net benefits of development in a flood plain and then looking for an optimal safety factor for the levee heightening strategy for a given time series of future flood events. The study then examines how the levee heightening strategy is affected by the magnitude of climate change.

Few comments are listed in the attached document.

Thank you for your comments

Sometimes in text and tables Thecnosociety is one word, others it is split. Please uniform

The same for Greensociety.

Done. Now all the times we use Thecnosociety or Greensociety.

Line 41: …literature (ANY REFERENCE?)

Done.

Line 168: write Flood Area Ratio (FAR) because it is the first time it is defined

Done.

Line 178: “d” is not defined anywhere

It is the d parameters previously defined. We have changed the redaction.

Line 249: capital T in the last word

Done.

Line 259: the shape

Done.

Line 275: write Intergovernal Panel on Climate Change (IPCC) because it is the first time it is defined

Done.

Line 312: In the high value case (FIGURE 3) the optimal…

Done.

Line 315: maybe Figure 4

Done.

Line 312: maybe Figure 2?

Done. It is Figure 3 in line 312 and Figure 2 in line 321.

Line 373: This paper shows how the outcomes of the socio-hydrology problem described in [1] can BE reframed

Done

Table 1: horizontal lines will make the table more readable. Put (μ) and “use” in the same line of the rest

Done

Table 7: put after reference in text            

The Table 7 is the result of calculation. This Table is referenced in text.

Reviewer 2 Report

The topic is very interesting. The paper is well organized and well written. In my opinion it can be published as is. I have only two comments:

- the novelty of the study is rather limited, as it is based on already known methodologies;

- Water is not the most suitable journal for the publication of this work, maybe. Other MDPI journals would probably be more suitable.

In any case, I recommend its publication, even if I invite the Academic Editor to consider my previous comments.

Author Response

REVIEWER 2

The topic is very interesting. The paper is well organized and well written. In my opinion it can be published as is. I have only two comments:

- the novelty of the study is rather limited, as it is based on already known methodologies;

- Water is not the most suitable journal for the publication of this work, maybe. Other MDPI journals would probably be more suitable.

In any case, I recommend its publication, even if I invite the Academic Editor to consider my previous comments.

Thank you for your comments.

We were invited to send a paper to the Water Journal. Before send the paper we consulted with the editor on this question, receiving an affirmative response.